

# Identification of anthropogenic debris in the stomach and intestines of giant freshwater prawns from the Trang River in southern Thailand

Kanyarat Tee-hor[1], Thongchai Nitiratsuwan[2] and Siriporn Pradit[1]

[1] Faculty of Environmental Management, Prince of Songkla University, Hat Yai, Thailand
[2] Faculty of Science and Fisheries Technology, Rajamangala University of Technology Srivijaya, Sikao, Thailand

Corresponding author
Siriporn Pradit,
siriporn.pra@psu.ac.th

## ABSTRACT

**Background**. Anthropogenic waste, especially microplastics, is becoming more prevalent in the environment and marine ecosystems, where it has the potential to spread through food chains and be consumed by humans. Southeast Asian countries are home to giant freshwater prawns, a common freshwater species that is eaten around the world. Microplastic pollution in river water, sediment, and commercially significant aquatic species such as fish and mollusks has been observed, yet few studies have been conducted on giant freshwater prawns in the rivers of southern Thailand, where microplastics may contaminate prawns via the food they ingest. The purpose of this research was to investigate the accumulation of anthropogenic material in the organs of river prawns (*Macrobrachium rosenbergii*).

**Methods**. Microplastics in the stomachs and intestines of giant freshwater prawns were the focus of this study. Samples were digested with 30 ml of 10% potassium hydroxide (KOH), heated for 5 min at 60 °C, and then digested at room temperature. The quantity, color, and appearance of microplastics were assessed using a stereomicroscope after 12 h. Furthermore, polymers were examined using a Fourier transform infrared spectrophotometer (FTIR). Microplastic counts were compared between sexes. A $T$-test was used to compare male and female microplastic counts in the stomach and intestine, and the Pearson correlation was used to compare the association between microplastic counts in the stomach and intestine and carapace length (CL), length of abdomen (LA), and body weight (BW) of male and female giant freshwater prawns. The threshold of significance was fixed at $p < 0.05$.

**Results**. Based on the study results, a total of 370 pieces of anthropogenic debris were discovered in the stomachs and intestines of both female and male prawns. The average number of microplastics per individual was $4.87 \pm 0.72$ in female stomachs and $3.03 \pm 0.58$ in male stomachs, and $1.73 \pm 0.36$ in female intestines and $2.70 \pm 0.57$ in male intestines. The majority of microplastics found in females were within the $<100$ μm range, while males contained microplastics in the range of 100–500 μm. Both male and female prawns contained fibers (72.70%) and fragments (27.30%). Various polymers were identified, including cotton, rayon, and polyvinyl chloride (PVC). The study also explored the relationship between carapace length, length of abdomen, body weight, stomach weight, and the number of microplastics. The findings reveal a significant association between the number of microplastics and stomach weight in male prawns

($R = 0.495; p = 0.005$). These findings provide alarming evidence of anthropogenic debris ingestion in prawns and raise concerns about the future effects of anthropogenic pollution on giant freshwater prawns.

## INTRODUCTION

The world is well aware of the devastating effect that plastic waste has on the ecosphere. Plastics are a popular material used to make a variety of goods (*Plastics Europe, 2018*). They are a form of synthetic polymer that has been widely employed due to their light weight, strength, durability, and low cost, as well as their ability to be molded into various shapes and sizes using contemporary manufacturing processes (*Bogusz & Oleszczuk, 2017*). Thailand is among the top six nations that dump the most plastic into the sea (*Jambeck et al., 2015*).

Microplastics are microscopic plastic particles that are produced as a byproduct of commercial product manufacturing due to the breakdown of larger plastics by physical, chemical, and biological processes (*Arthur, Baker & Bamford, 2009*). These processes produce macroplastic (more than 25 mm), mesoplastic (5–25 mm), and microplastic (less than five mm) particles (*Desforges et al., 2014*), which are derived from primary sources of microplastics such as plastic beads from plastic manufacturers, microbeads in cosmetics, and fishing net fibers, as well as secondary sources of microplastics (*GESAMP, 2016*). Microplastic pollution in the environment causes microplastics to infiltrate the food chain, where they can directly impact organisms and ecosystems (*Lusher, Hollman & Mendoza-Hill, 2017*). It can now be detected in a variety of ways. It has been documented that many forms of microplastics have been ingested by zooplankton, shrimp, and animals living in alluvium or mangrove soil (*Pradit et al., 2021*; *Abbasi et al., 2018*; *Devriese et al., 2015*; *Moore et al., 2001*; *Murray & Cowie, 2011*). Several studies have shown that consuming too much microplastic-contaminated food on a regular basis increases the likelihood of acquiring allergies (*Pironti et al., 2021*). Microplastics can obstruct the activity of organs in the body, such as the circulatory system, because they are small enough to enter the bloodstream, causing pain and irritation to internal organs. They can also enter the digestive system where they can cause gastric cancer. The most dangerous effect of microplastics on the body is genetic mutation (*Thushari et al., 2017*).

South and Southeast Asia, in addition to some parts of the Pacific Islands, are home to giant freshwater prawns (*Petcjun & Siriwat, 2016*). Giant freshwater prawns are large shrimp that live in freshwater waterways along rivers and canals, and are usually observed in regions where the water is flowing and clean. They are commonly consumed both locally and internationally because the flesh is excellent and has a high nutritional value. Because of their high price, the species is popular among fishermen (*Nitiratsuwan et al., 2022*). Fertilized female giant freshwater prawns move to the river mouth or brackish water to

spawn during the breeding season before moving back to fresh water. The feeding habitats of prawns involve consuming a wide variety of organic material (*Sitthi, 2011*). Microplastics are abundant in river water and soil, and aquatic species such as prawns may absorb them while feeding. Microplastic consumption has been researched in different shrimp species, such as *Paratya australiensis*, and were found in 36% of the shrimp, with an average of $0.52 \pm 0.55$ items/ind ($24 \pm 31$ items/g) (*Nan et al., 2020*.). The gut of *Nephrops norvegicus* was investigated, and 83% of the animals analyzed had plastics (mostly filaments) in their stomachs (*Murray & Cowie, 2011*). There has been little academic research on this topic in Thailand, and there have been no reports of microplastic buildup in the Trang River. This study investigates the presence of anthropogenic waste, such as microplastic-like debris, in the gastrointestinal tracts of giant freshwater prawns.

## MATERIALS & METHODS

### Sample collection and preparation

In September of 2022, a total of 6 kg of giant freshwater prawns (*M. rosenbergii*) was purchased randomly from coastal fishermen who operate on the Trang River in the Trang province (Fig. 1). To conduct a microplastic analysis, a total of 60 giant freshwater prawns were randomly selected, with 30 males and 30 females. Male and female characteristics of giant freshwater prawns are shown in Fig. 2. The sample size used in this investigation is consistent with previous studies conducted by *Cole et al. (2013)*, *Pradit et al. (2020)*, and *Jitkaew et al. (2023)*. It is important to note that this species is commonly consumed in Thailand. To preserve the giant freshwater prawn samples, they were carefully wrapped in aluminum foil and stored in a freezer at a temperature of $-20\,°C$ in preparation for further analysis.

### Prevention of microplastic contamination

A blank test was performed using a 250-ml beaker filled with distilled water and placed in a laboratory. After 24 h, the distilled water in the beaker was filtered using filter paper, oven-dried, and examined under a microscope to ensure the absence of microplastics. The experiment took place in a clean room with a fume hood, and no disturbances such as wind were present in the laboratory. The researcher wore gloves, a gown, and a surgical cap throughout the experiment. During the lab analysis, aluminum foil was placed over the glass beaker containing the dissected sample (*Pradit et al., 2023*). To minimize the impact of exogenous microplastics, no plastic instruments were used on the samples during the experiment (*Pan et al., 2021*). All materials were cleaned and rinsed with distilled water before use.

### Anthropogenic debris identification

The frozen giant freshwater prawn samples (*M . rosenbergii*) were defrosted at room temperature. Carapace length (CL) and length of abdomen (LA) were measured in centimeters (Fig. 3), and body weight (BW) was measured in grams, in accordance with Food and Agriculture Organization of the United Nations (FAO) guidelines. The description from *Dehaut et al. (2016)* served as the basis for the sample analysis procedure.

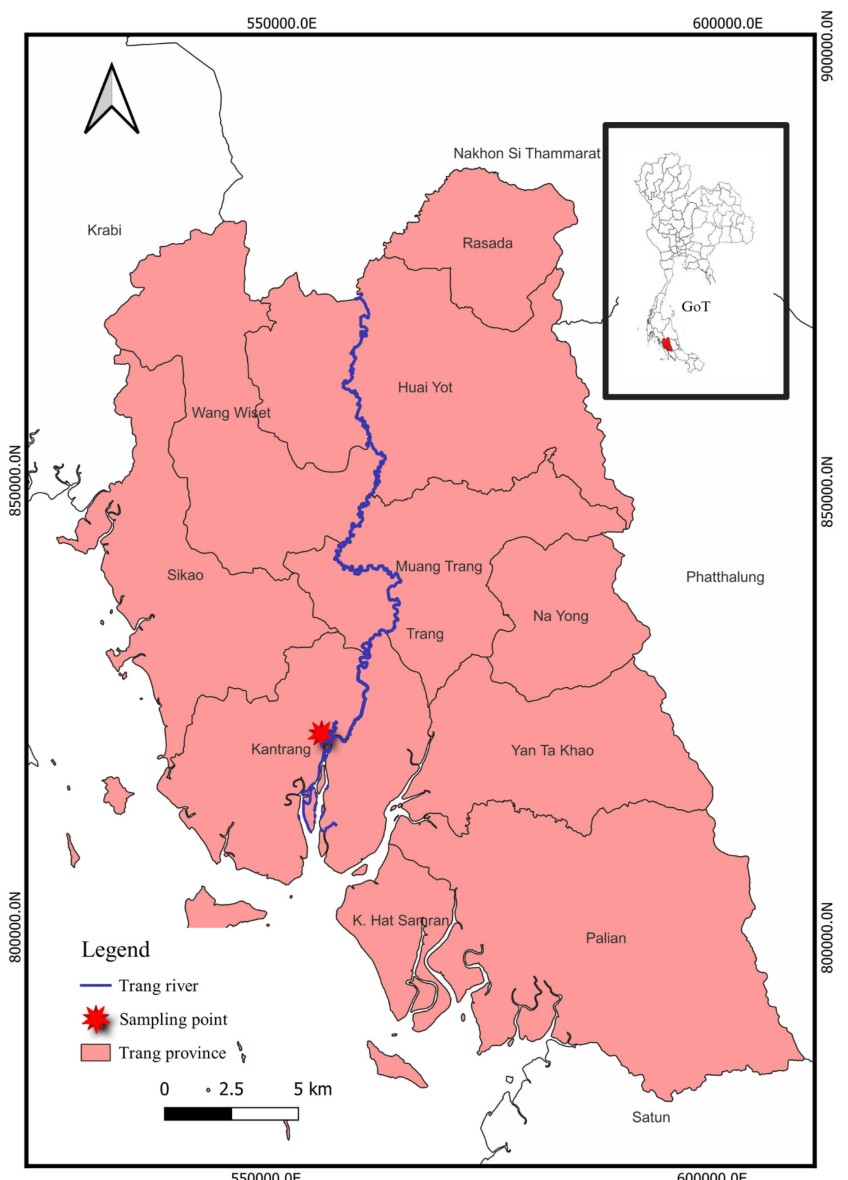

**Figure 1** **The study area of the Trang River, Trang Province, Thailand.**

The samples' intestines and stomachs (Fig. 2) were removed using thin forceps, cut into small pieces, and placed in a beaker. The alkaline technique was applied to digest the dissected stomachs and digestive tracts of the samples (*Cole et al., 2014*; *Ding et al., 2018*). The samples were then placed in 30 ml of 10% potassium hydroxide (KOH) solution, stirred continuously for 1 min with a stirring rod, covered with aluminum foil to prevent foreign matter contamination from the air, heated to 60 °C for 5 min, and left to degrade for another 12 h at room temperature. The samples were then filtered with a 20-micron filter cloth. The filter cloth (new, made of nylon) was dried in a hot air oven at 50 °C for 5 h. Studies conducted after digestion can benefit from density separation. The primary
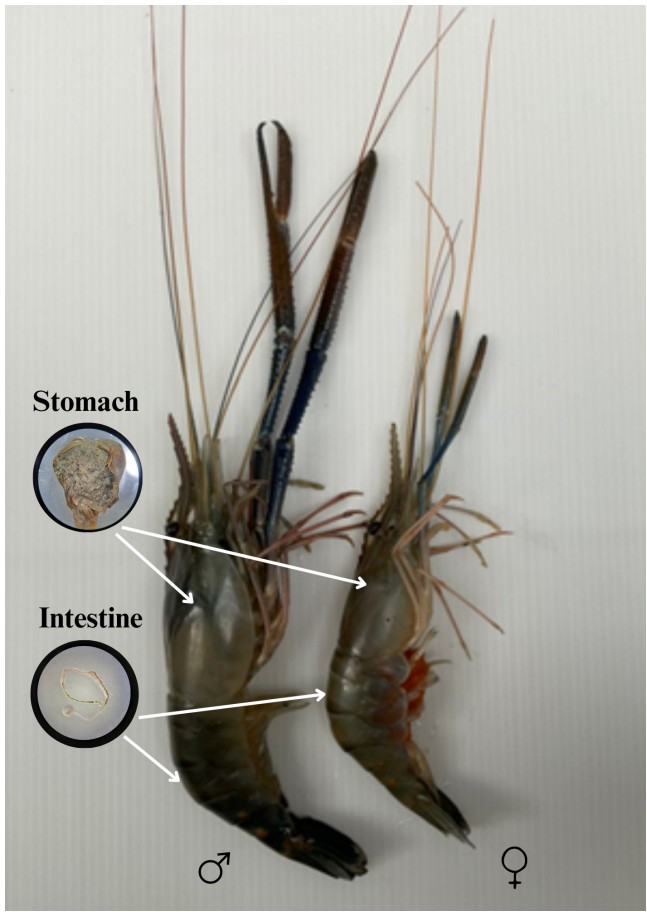

**Figure 2    Male and female characteristics of giant freshwater prawns.**

aim is to separate microplastics from sediment or other inorganic material that was not dissolved during enzymatic or chemical digestion (*Stock et al., 2019*), and when there is a significant amount of inorganic material present (*Lusher et al., 2017*). Because there was no organic or inorganic material left (such as sand or chitin) after digestion with KOH (10%) in our investigation, the density separation step was skipped. The method of using alkaline digestion was adapted for the dissolution of the biota of invertebrates and fish and has proven largely efficacious in removing biogenic material (*Lusher et al., 2017*). In the absence of debris, organic matter, shells, or cartilage, which can prevent the identification of microplastics on the filter, an alkaline digestion was deemed to be efficient (*Dehaut et al., 2016*).

The microplastic samples on the filter cloth were carefully counted and their sizes measured. Additionally, their characteristics and color were observed using an Olympus SZ61 three-dimensional viewing system equipped with a light-emitting diode. The microplastics were counted as individual pieces. The size of the microplastics was categorized into four classes: <100 μm; 101–500 μm; 501–1,000 μm; and >1,000 μm. Furthermore, the types of microplastics were classified into two categories: fibers and
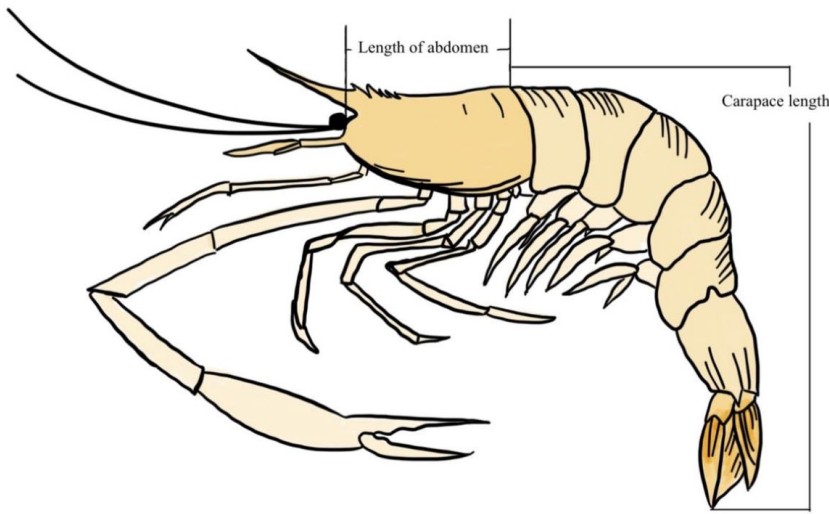

**Figure 3  Carapace length and length of the abdomen of a giant freshwater prawn.**

fragments. Randomly selected microplastics longer than 100 μm were analyzed on a Fourier transform infrared spectrophotometer (FTIR), using the Attenuated Total Reflectance mode to identify their composition; Spectrum Two; Perkin Elmer Spectrum IR version 10.6.2, spotlight 200i; Perkin Elmer, Seer Green, UK. In the study, the wavelength spanned from 4,000 cm$^{-1}$ to 400 cm$^{-1}$. The acquired spectrum was compared to the standard library spectrum.

## Data analysis

Descriptive data on the number, size, color, and shape of microplastics was collected in Microsoft Excel (Office Professional Plus 2019). Data was presented in the form of a mean standard error. The t-test was used to compare the number of microplastics found in the intestines and stomachs of male and female giant freshwater prawns. The relationship between the number of microplastics in the intestines, stomachs, carapace length, length of the abdomen, and body weight between male and female giant freshwater prawns was measured using the Pearson correlation. The significance level was set at $p < 0.05$.

## RESULTS

### Abundance of anthropogenic debris in the stomach and intestine of giant freshwater prawns

Sixty giant freshwater prawns (30 females and 30 males) were tested. Each giant freshwater prawn was measured for CL, BW, LA, and SW before the analysis was conducted (Table 1). The number of microplastics in the stomachs and intestines of female and male giant freshwater prawns were 4.87 ± 0.72 items/individual, 1.73 ± 0.36 items/individual, 3.03 ± 0.58 items/individual and 2.70 ± 0.57 items/individual, respectively (Table 2). The number of microplastics in the stomachs of female giant freshwater prawns and male giant freshwater prawns ($p = 0.866$) and intestines ($p = 0.171$) was not statistically different.

Tee-hor et al. (2023), *PeerJ*, DOI 10.7717/peerj.16082

**Table 1  Carapace length (cm), weight (g), length of abdomen (cm), and stomach (g) in giant freshwater prawns.**

| Sex | Carapace length (cm) | | | Weight (g) | | | Length of abdomen (cm) | | | Stomach (g) | | |
|---|---|---|---|---|---|---|---|---|---|---|---|---|
| | max | min | mean ± SE | max | min | mean ± SE | max | min | mean ± SE | max | min | mean ±SE |
| Female ($n = 30$) | 5.40 | 3.40 | 4.41 ± 0.08 | 99.11 | 28.94 | 54.83 ± 2.96 | 10.00 | 7.00 | 8.49 ± 0.13 | 1.28 | 0.22 | 0.56 ± 0.04 |
| Male ($n = 30$) | 7.00 | 3.60 | 5.15 ± 0.17 | 175.61 | 33.06 | 89.07 ± 8.07 | 11.40 | 7.00 | 9.26 ± 0.19 | 6.33 | 0.30 | 1.39 ± 0.24 |

Table 2 Anthropogenic debris abundance in giant freshwater prawns.

| Sex | Body | Microplastics item | |
| --- | --- | --- | --- |
| | | Total Microplastics | Average item/individual |
| Female | Stomach | 146 | $4.87 \pm 0.72$ |
| | Intestine | 52 | $1.73 \pm 0.36$ |
| Male | Stomach | 91 | $3.03 \pm 0.58$ |
| | Intestine | 81 | $2.70 \pm 0.57$ |

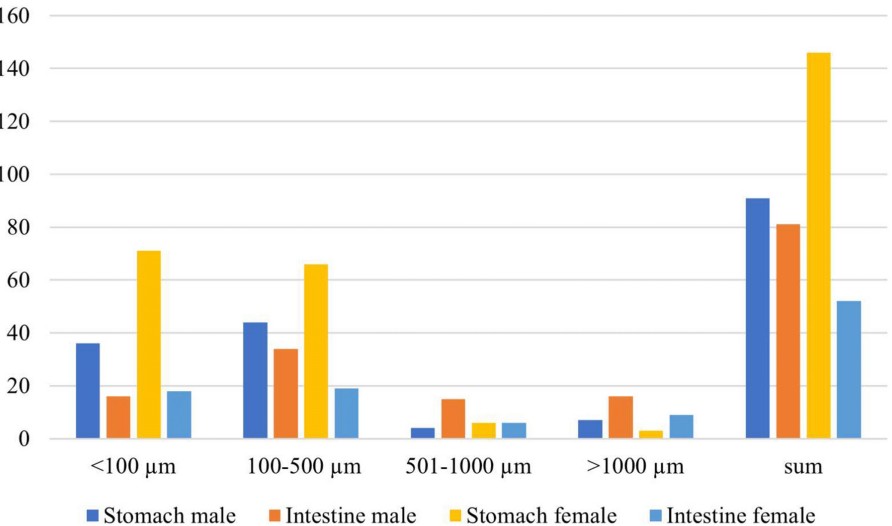

Figure 4 Size of anthropogenic debris in female and male giant freshwater prawns.

## Anthropogenic debris size

In the stomachs of female giant freshwater prawns, the most common size of microplastics found was <100 μm whereas in the stomachs of male giant freshwater prawns, the most common size of microplastics found was 100–500 μm. In the intestines of female and male giant freshwater prawns, the most common size of microplastics found was 100–500 μm. See more details in Fig. 4.

## Anthropogenic debris type, color, and polymer

The intestines of male giant freshwater prawns included 91.36% fiber-type microplastics, followed by the stomachs (85.71%) of male giant freshwater prawns and the intestines (69.23%) and stomachs (55.48) of female giant freshwater prawns. Fragment-type microplastics were found in 44.52% of female giant freshwater prawn stomachs, followed by female giant freshwater prawn intestines (30.77%), and male giant freshwater prawn stomachs (14.25%) and intestines (8.64%) (Table 3). The forms and sizes of the microplastics differed.

Blue (61.35%), black (32.70%), red (5.68%), and yellow (0.27%) microplastics were found. Blue was the most prevalent hue discovered in the stomach and intestines of both

**Table 3  Anthropogenic debris type and color in female and male giant freshwater prawns.**

| Category of microplastics | | *Macrobrachium rosenbergii* | | | |
|---|---|---|---|---|---|
| | | **Female** | | **Male** | |
| | | **Stomach** | **Intestine** | **Stomach** | **Intestine** |
| Type (%) | Fiber | 55.48 | 69.23 | 85.71 | 91.36 |
| | Fragment | 44.52 | 30.77 | 14.29 | 8.64 |
| Color (%) | Black | 37.67 | 16.00 | 27.18 | 44.44 |
| | Blue | 57.53 | 78.00 | 67.03 | 50.62 |
| | Red | 4.79 | 6.00 | 8.79 | 3.70 |
| | Yellow | 0.00 | 0.00 | 0.00 | 1.23 |

**Table 4  The relationship between size of giant freshwater prawns and the amount of anthropogenic debris in stomach (ST) and the intestines (IN).**

| | | CL (cm) | AL (cm) | SW (g) | BW (g) | ST (items) | IN (items) |
|---|---|---|---|---|---|---|---|
| *Macrobrachium rosenbergii* (female) | CL (cm) | 1 | .805[**] | .385[*] | .840[**] | .292 | −.176 |
| | AL (cm) | | 1 | .468[**] | .894[**] | −.062 | −.122 |
| | SW (g) | | | 1 | .321 | .109 | −.074 |
| | BW (g) | | | | 1 | .222 | −.178 |
| | ST (items) | | | | | 1 | .083 |
| | IN (items) | | | | | | 1 |
| *Macrobrachium rosenbergii* (male) | CL (cm) | 1 | .912[**] | .371[*] | .932[**] | .029 | −.045 |
| | AL (cm) | | 1 | .352 | .903[**] | .121 | −.145 |
| | SW (g) | | | 1 | .407[*] | .241 | .495[**] |
| | BW (g) | | | | 1 | .085 | −.119 |
| | ST (items) | | | | | 1 | .144 |
| | IN (items) | | | | | | 1 |

female and male giant freshwater prawns, followed by black, red, and yellow. Yellow microplastics were not found in the stomachs and intestines of female giant freshwater prawns nor male giant freshwater prawns (Table 3). Cotton (70.37%), rayon (25.93%), and polyvinyl chloride (PVC) (3.70%) were the polymer types found in *M. rosenbergii* (Fig. 5).

**Correlation between the size of giant freshwater prawns and anthropogenic debris in the stomach and intestines**

The study examined the relationship between CL, LA, BW, and stomach weight, and the number of microplastics. The findings revealed a significant association between the number of microplastics and stomach weight in male prawns ($R = 0.495$; $p = 0.005$) (Table 4).

# DISCUSSION

This study confirmed that microplastics were detected in the stomach and intestines of female and male giant freshwater prawns. More microplastics were found in the stomach than in the intestines in both female and male giant freshwater prawns; $4.87 \pm 0.72$

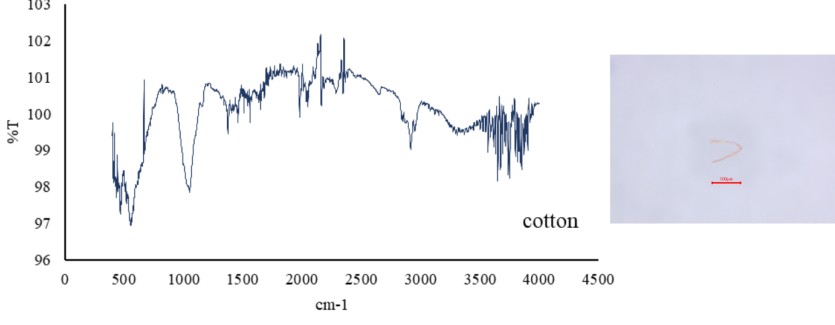

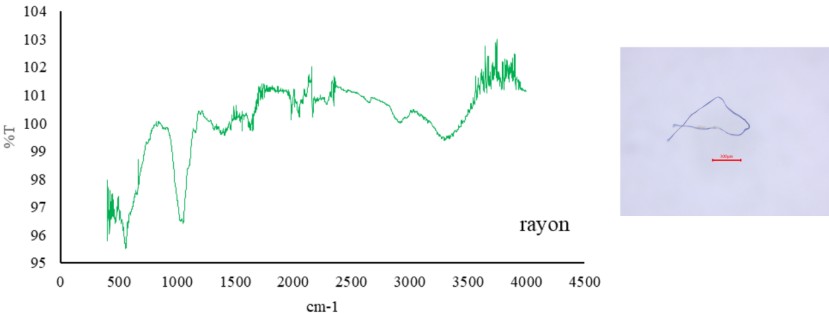

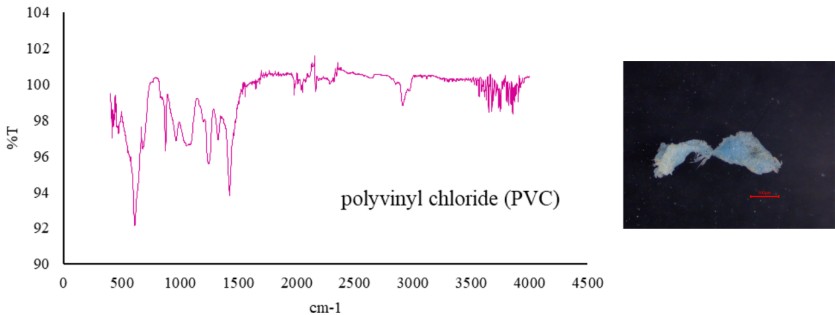

**Figure 5 Polymer found in giant freshwater prawns.**

items/individual, $1.73 \pm 0.36$ items/individual, $3.03 \pm 0.58$ items/individual, $2.70 \pm 0.57$ items/individual, respectively. There was no significant difference in the accumulation of microplastic particles in the stomachs ($p = 0.866$) and intestines ($p = 0.171$) of female and male giant freshwater prawns. This is consistent with a study conducted in shrimp, *Crangon* crangon, that found microplastics were most common in the gastrointestinal tract (*Devriese et al., 2015*). The number of microplastics found in each type of prawn and shrimp depends on the environment where the prawn samples are selected, such as rivers, aquaculture ponds, lakes, and seas. When comparing the present study with other studies (Table 5), the number of microplastics detected in this study varied by location. The amount of microplastics found in this study is similar to the results obtained by *Goh et*

*al. (2022)*, but less than other studies in Thailand (*Jitkaew et al., 2023*; *Reunura & Prommi, 2022*; *Pradit et al., 2021*). This could be because there are fewer sources of microplastics in the aquatic environment from this study (south-west Thailand) than in the areas from other studies (south-east Thailand). The most common size of microplastics found in this research was 100–500 µm (44.05%), followed by <100 µm (38.11%), >1,000 µm (9.46%), and 501–1,000 µm (8.38%), respectively. The size of the microplastics found is likely related to their toxicity. Smaller microplastics can better absorb hydrophobic materials from a production process or from the environment, resulting in humans being at a greater risk of exposure to toxic chemicals (*Lusher, Hollman & Mendoza-Hill, 2017*). Microplastics can absorb the additional chemicals (plastic additives) used in the manufacturing process that give plastic products their color and characteristics (*Pradit et al., 2020*). The results of interactions between selected microplastics and heavy metals strongly support the hypothesis that microplastics can absorb heavy metals and act as a vector for heavy metal ion distribution in the marine ecosystem (*Goh et al., 2022*). As a result, microplastics appear to be poison transporters for aquatic creatures that consume microplastics. Heavy metals can accumulate in marine creatures, increasing in concentration over time. This concentration provides a record of the availability of metal species in the environment (*Rainbow, 2002*).

The most abundant category of microplastics found was fibers, followed by fragments, which is consistent with the findings of several studies (*Pradit et al., 2021*; *Goh et al., 2021*; *Gurjar et al., 2021*; *Hossain et al., 2020*; *Devriese et al., 2015*). According to *Weinstein, Crocker & Gray (2016)*, it was reported that polymer fibers can float on water for a long time due to their low density, while fragments with rough surfaces are broken down by natural forces. The fibers found in this study likely originated from floating fibers in river water, and the fiber strands of polymer most likely came from fishing nets and clothing lint (*De Witte et al., 2014*).

The microplastics found in both male and female giant freshwater prawns were blue, black, and red, while yellow microplastics were only found in the intestines of male giant freshwater prawns. This is similar to the results of previous studies (*Jitkaew et al., 2023*; *Reunura & Prommi, 2022*; *Pradit et al., 2021*; *Goh et al., 2021*; *Gurjar et al., 2021*; *Nan et al., 2020*). It was also found that plastics with a long lifespan and darker colors are more likely to be contaminated with other chemical substances than long-lived lighter-colored plastics (*De Witte et al., 2014*). *Wright, Thompsom & Galloway (2013)* reported that living organisms choose to eat plastics that look similar to their regular food, causing them to acquire microplastics in their gastrointestinal tract. According to *Sitthi (2011)*, giant freshwater prawns eat all types of food, both living and nonliving, including fish, seedlings, and other prawns.

In this study, three polymer types were found in giant freshwater prawns. The results reveal that natural polymer cotton was the most abundant, followed by semi-synthetic polymer (rayon) and synthetic polymer (PVC), respectively. The use of detergent in laundering likely results in increased microfibers (*Zambrano et al., 2019*) which are then suspended and accumulate in bottom sediment or in water currents (*Henry, Laitala & Klepp, 2019*). This study found polyvinyl chloride (PVC) in the stomachs of male giant
**Table 5** Microplastic abundance in giant freshwater prawns.

| Shrimp | Location | Method | Abundance of microplastics | Color | Shape | Size | Polymer | References |
|---|---|---|---|---|---|---|---|---|
| Metapenaeus moyebi (n = 17) | Khlong U-Taphao, Songkhla | $H_2O_2$ 30% | 14.76 ± 1.98 items/ individual | blue, black, other | fiber, fragment | less than 100 μm | rayon, polyester, PET, PP, Poly (Ethylene:Propylene) | Jitkaew et al., 2023 |
| Macrobrachium rosenbergii (n = 17) | | | 11.24 ± 1.74 items/individual | | | larger than 1,000 μm | | |
| Litopenaeus vannamei (n = 150) | | | 11.00 ± 4.60 items/ individual | | | | | |
| Macrobrachium rosenbergii (n = 300) | Thailand | $HCO_2K$ 99% | 33.43 ± 19.07 items/ individual (male) | black, red, white, blue, yellow, green | fiber, fragment, film, spheres | 500–1,000 μm | PE, polycaprolactone, polyvinyl alcohol, acrylonitrile butadiene styrene | Reunura & Prommi (2022) |
| | | | 33.31 ± 19.42 items/ individual (female) | | | | | |
| Parapenaeopsis hardwickii (n = 18) | Songkhla Lake, Southern Thailand | KOH 10% | 4.11 ± 1.12 pieces/stomach | black, red blue, white | fiber | 500–1,500 μm | rayon, polyester, polyvinyl alcohol, PE, paint | Pradit et al. (2021) |
| Metapenaeus brevicornis (n = 18) | | | 3.78 ± 0.48 pieces/stomach | | | 500–5,000 μm | | |
| Metapenaeus elegans (n = 20) | Songkhla Province, Southern Thailand | KOH 10% | 3.70 ± 1.12 number of MPs/individual | black, red blue, gray, transparent | fiber | 150–3,800 μm | PE | Goh et al. (2021) |
| Fenneropenaeus indicus (n = 20) | | | 3.45 ± 0.04n/individual | | | | | |
| Metapenaeus monoceros (n = 60) | | | 7.23 ± 2.63 items/individual 78.48 ± 48.37 MPs/gram of the gut material | | | | | |
| Parapeneopsis stylifera (n = 50) | North Eastern Arabian sea | $HNO_3$ 69% | 5.36 ± 2.81 items/individual 64.79 ± 24.58 MPs/gram of the gut material | blue, translucent, black, red | fiber, fragment, pellet, film, beads | <100 μm, -greater than 1,000 μm | PE, PP, PA, nylon, PES, PET | Gurjar et al. (2021) |
| Penaeus indicus (n = 70) | | | 7.40 ± 2.60 items/individual 47.5 ± 38.0 MPs/gram of the gut material | | | | | |
| P. monodon / gastrointestinal tract (n = 50) | Northern Bay of Bengal | $H_2O_2$ 30% | 6.60 ± 2.00 pieces/gram | blue, black, transparent, green, red | fiber, fragment | 250–5,000 μm | rayon, polyamide | Hossain et al. (2020) |
| M. monocerous / gastrointestinal tract (n = 100) | | | 7.80 ± 2.00 pieces/gram | blue, black, transparent, green | | <250–5,000 μm | | |
| Fenneropenaeus indicus (n = 330) | coastal waters off Cochin, Kerala, India | KOH 10% | 0.39 ± 0.60 microplastics/gram | red, blue, black, transparent, green | fiber, fragment | 157–2,785 μm | polyamide, polyester, polyethene, PP | Daniel, Ashraf & Thomas (2020) |
| Paratya australiensis (n = 100) | Victoria, Australia | NaOH | 0.52 ± 0.55 pieces/ individual | black, red gray, white blue, green, transparent, yellow | fiber, fragment, film, pellet | 36–4,668 μm | rayon, polyester, polymide | Nan et al. (2020) |
| Crangon crangon (n = 165) | North sea | HNO3: HClO4 4:1 | 1.23 ± 0.99 items/ individual | transparent, translucent, orange, yellow-greenish, purple-blue, pink | fiber | 200–1,000 μm | – | Devriese et al. (2015) |
| Macrobrachium rosenbergii (n = 60) | Thailand | KOH 10% | female; stomach 4.87 ± 0.72 MPs/individual intestine 1.73 ± 0.36 MPs/individual | black, red, blue, yellow | fiber, fragment | | cotton, rayon, PVC | This study |
| | | | male; stomach 3.03 ± 0.58 MPs/individual intestine 2.70 ± 0.57 MPs/individual | | | | | |

freshwater prawns, similar to a study on *Litopenaeus vannamei* in the Korean Sea (*Yoon et al., 2022*), which found that the PVC likely came from food packaging and fishing equipment. The study of the correlation between microplastic content and CL, LA, BW, and stomach weight found that there was no correlation between female giant freshwater prawns and microplastic content in the stomach and intestines, while there was a significant correlation between male giant freshwater prawns and intestinal microplastic content and stomach weight at the level of $R = 0.495$; $p = 0.005$. This indicates that the high gastric weight of giant freshwater prawns may result in an increase in intestinal microplastic content in proportion to the stomach. CL, LA, and BW were not associated with the number of microplastics in female and male giant freshwater prawns.

It is projected that the problem of plastic waste will worsen due to the excessive use and consumption of single-use plastics (*Silva et al., 2021*) as well as an increase in the demand for personal protective equipment (PPE) such as masks, and rubber gloves, which will lead to an increase in PPE waste (*Okuku et al., 2021*). A public awareness campaign aimed at changing people's attitudes regarding the environment is critical (*Sornplang et al., 2022*). Diffusion can occur when microplastics are smaller than five mm, causing widespread pollution of the environment. If an organism is exposed to this environment for a prolonged period of time, there is a greater chance that the exposure will have negative effects. These effects could include obstructions in the gastrointestinal tract of organisms, increased mortality rates, decreased ability to reproduce, and inhibition of metabolism. However, depending on the size, shape, and type of contaminated plastic in the environment, as well as the quantity and concentration discovered (*Cole et al., 2013*; *Zhang et al., 2017*), other hazardous additive contaminants may be released which could serve as an intermediary to other pollutants, further harming aquatic animals and humans.

## CONCLUSIONS

In this study, anthropogenic waste was discovered in the stomachs and intestines of giant freshwater prawns (*M. rosenbergii*). This discovery indicates that microplastic pollution, which is caused by a range of human activities, is harmful because microplastics can enter the food chain. Fibers were the most prevalent category of microplastic found in prawn organs. Blue, black, and red microplastics were identified in the intestines of both male and female giant freshwater prawns, whereas yellow microplastics were found in the intestines of male giant freshwater prawns. Cotton, rayon, and PVC were also discovered in these giant freshwater prawns. Although microplastics are evacuated with waste, some persist in the tissue. Consequently, to reduce plastic pollution in the seas in the future, people need to be informed of the government's management and act immediately to remedy issues with waste disposal.

### Funding

This work supported a research grant from the Coastal Oceanography and Climate Change Research Center (COCC) and a graduate scholarship from the Faculty of Environmental Management. Academic Year 2022, Prince of Songkla University and thanks to Kivita Chandran, Kasvinraj Murugiah, Hemaadarshini Krebanathan for helping in the analysis of the samples. The funders had no role in study design, data collection and analysis, decision to publish, or preparation of the manuscript.

### Grant Disclosures

The following grant information was disclosed by the authors:
Coastal Oceanography and Climate Change Research Center (COCC).
Faculty of Environmental Management.
Prince of Songkla University.

### Competing Interests

The authors declare there are no competing interests.

### Author Contributions

- Kanyarat Tee-hor performed the experiments, analyzed the data, prepared figures and/or tables, authored or reviewed drafts of the article, and approved the final draft.
- Thongchai Nitiratsuwan conceived and designed the experiments, analyzed the data, prepared figures and/or tables, and approved the final draft.
- Siriporn Pradit conceived and designed the experiments, performed the experiments, analyzed the data, authored or reviewed drafts of the article, and approved the final draft.

### Data Availability

    The raw data are available in the Supplemental Files.

### Supplemental Information

Supplemental information for this article can be found online at http://dx.doi.org/10.7717/peerj.16082#supplemental-information.

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
