# Peer review of "Identification of anthropogenic debris in the stomach and intestines of giant freshwater prawns from the Trang River in southern Thailand"

_PeerJ, doi:10.7717/peerj.16082_

## Round 0.1 · original submission · Major Revisions

Although the authors see merit in your paper, they have raised a number of concerns that should be addressed in your paper before acceptance for publication. The language used in the paper should be improved, and reviewer 1 has tried to point out instances where that should be done. However, this is not comprehensive, and the authors should make sure the language used throughout the paper is clear and intelligible.

Reviewer 1 recommends additions to be made in the introduction on the consumption of microplastics by prawns. Under the methods section, you also need to elaborate on the contamination controls you employed.

The discussion on plastics should be improved and placed in a wider context, not just focusing on prawns and the local context.

The conclusions should also be strengthened by highlighting the major findings and providing direction for future research. All minor and major comments raised by the reviewers should be addressed in the revised manuscript.

·

Basic reporting

The authors investigate an issue of microplastic contamination in river prawns in Thailand. This is an important issue worth examining as microplastics are widespread pollutants with many effects and implications on ecosystems and human health. The use of English could be improved, with simple grammatical mistakes that could be fixed easily. This article is of a straightforward objective which can be greatly improved if many details are added in.

Experimental design

Many details need to be added in- especially in the methods section on contamination controls. More details are found in the comments section.

Validity of the findings

The findings are valid, however when reporting the results the authors need to add more details.

The discussion is also a little short, and could be extended with greater elaboration in relation to the effects of MPs in the wider context.

Additional comments

From the background (line 46)
The authors state the problem of microplastics ( in general) but the problem of prawns consuming microplastics is not stated obviously. The authors need to make it clear in the background why they are investigating prawns in relation to microplastics.

Line 63- replace anthropogenic debris with microplastics if this is what you mean- there are many other types of debris, which are not microplastics.

Line 65- what do the results here mean? The average number based on colour? Or based on the previous sentence about the various types of stomachs? The authors should state the breakdown in % of the types of microplastic types, not just the major (fiber) found. The organisation of results in this section from lines 63-72 need to be structured better.

Line 75- change harms to harm

Line 95- please remove the statement comparing MPs to bacteria and viruses

Line 97- missing reference at the end of sentence

Lines 101-106- The listing of locations is very extensive and lengthy, kindly shorten

The authors mention the problem of MPs in aquatic ecosystems but need to draw the direct link between this contaminant and the prawns consuming this before jumping to line 112. How do prawns consume MPs? This is missing in the introduction.

Line 131- Many details are missing in the section of contamination control. How did you not detect any microplastics from the beaker? By microscopic observation with a rafter? Was the experiment performed in a clean room/ film hood? And what type of lab coat, cotton or? Also, was glassware/stainless steel ware used for the experiment? Many details are missing (same comment for section on sample collection and preparation)

Line 141- intestines were removed using what tools?

Line 145- What material was the filter cloth?

Lines 171-175- too much listing of data, which is just needed to be referenced in a table/figure. Same comment for section (anthropogenic debris: size).

Line 203- what is the % of each colour detected?

Line 207- same comment for the polymer type

Line 221- change the phrase” statistical signifcance” to “no significant difference”

Line 232- why is this phenomenon so? Can you relate it to the surrounding area/ pollution levels/ hydrodynamic conditions?

Line 233-234- what are the %s for size?

Line 236- can elaborate more about the point on toxic chemicals for better depth of discussion

·

Basic reporting

This paper investigated microplastic contamination in the stomach and intestines of giant freshwater shrimp from the southern rivers of Thailand.
Overall, the importance of this paper is sufficient as the first result of confirming microplastics in shrimp in the Trang River, giving examples of the harmfulness of microplastics and examples of microplastics in shrimp.

In conclusion, there is no problem with this paper being published in PeerJ after undergoing minor revisions.

Experimental design

Experiments were conducted on 30 female shrimp and 30 male shrimp, and the number of samples was sufficiently secured without any major problems. Pretreatment of shrimp intestines was carried out with 10% KOH, but it is necessary to confirm the density separation. It is also necessary to check whether the decomposition of organic matter is complete.
It is considered that there is no big problem in analyzing microplastics using ATR of FTIR.

Validity of the findings

The number, size, color, and type of microplastics in shrimp were investigated, and the investigated values were confirmed as meaningful values.
In the discussion, the number of microplastics was compared with other studies, and a correlation with existing studies was found, so it can be said to be reliable.

Additional comments

The conclusions are summarized too briefly, making it difficult to establish the results, importance, and direction of future research.

---

## Round 0.2 · Minor Revisions

Although the revised draft has significantly improved and you have addressed most of the comments raised by reviewers, Reviewer 2 has lingering concerns that you need to address. In your methods, you need to provide more details on the pretreatment process used in microplastic analysis. You also need to strengthen the discussion with details on organic matter decomposition rates and microplastic separation methods, and how these potentially influence your findings and conclusions drawn.

·

Basic reporting

The authors have responded to all my comments and the paper is now suitable for publication.

Experimental design

Methods are improved.

Validity of the findings

The results have been better discussed after revision.

·

Basic reporting

This paper has been generally revised for the points pointed out in the first review, but more explicit revisions are needed.

Experimental design

Since the method for density separation after decomposition of organic matter is not used, it is necessary to develop a method for separating microplastics.
The time for complete organic degradation is relatively short, so clear information on whether complete organic degradation has been achieved is needed.

Validity of the findings

Regarding microplastics in shrimp, it is considered that the color, size, characteristics, etc. are reasonable by comparing with various data.

Additional comments

Overall, there are no major problems in the paper, but since the pretreatment process is important in microplastic analysis, it is necessary to supplement with clearer methods, organic matter decomposition information, and microplastic separation methods.

---

## Round 0.3 · Minor Revisions

While your manuscript is much improved and you have satisfactorily addresses comments reraised by all reviewers, it still needs further improvement before it is accepted for publication. There are a number of instances where the grammar needs improvement. Also, make efforts to eliminate typos from your manuscript. Acceptable manuscripts should be clear and free of typos and errors. Some of the corrections, include:
Ln 47: 'sediments'
Ln 64: delete 'found to be'
Ln 64& 65; differrentiate between stomachs and intestines in the methods
Ln 68: is cotton a microplastic? Your title and rest of the manuscript focuses more on microplsatics
Ln 69: This sentence describes methods and should be moved to the methods section.
Ln 154: Separate - temperature.The
Ln 155: Separate - 50℃for

This is just a snapshot of the lack of clarity and typos in your manuscript. So, I invite you to thoroughly go through your manuscript and improve clarity, grammar and eliminate all typos/errors.

---

## Round 0.4 · Minor Revisions

I appreciate your responses to my comments, but these were not exhaustive. I only provided a few comments as a guide to the extensive corrections you need to make to clear errors (typos and grammar) in your manuscript and improve clarity. I need to see a more comprehensive revision of your manuscript before it is accepted for publication. If possible, you can enlist the help of a proficient speaker to improve your grammar.

**Language Note:** The Academic Editor has identified that the English language must be improved. PeerJ can provide language editing services - please contact us at copyediting@peerj.com for pricing (be sure to provide your manuscript number and title). Alternatively, you should make your own arrangements to improve the language quality and provide details in your response letter. – PeerJ Staff

---

## Round 0.5 · accepted · Accept

I had recommended substantial editing of the revised draft of the manuscript (ms) to eliminate errors and typos and address any grammatical errors. Given the comprehensive revision that has been done, I recommend this version of the ms for acceptance. The authors had already comprehensively addressed the minor comments raised by the reviewers. The current version of the ms is ready for publication. Make sure that the final ms submitted is the latest version without the track changes.